# Potential Impact of a Pregnant Woman’s Microbiota on the Development of Fetal Heart Defects: A Review of the Literature

**DOI:** 10.3390/biomedicines12030654

**Published:** 2024-03-14

**Authors:** Katarzyna Zych-Krekora, Oskar Sylwestrzak, Michał Krekora, Przemysław Oszukowski, Mariusz Grzesiak

**Affiliations:** 1Department of Perinatology, Obstetrics and Gynaecology, Polish Mother’s Memorial Hospital Research Institute, 93-338 Lodz, Poland; mariusz.grzesiak@gmail.com; 2Department of Obstetrics and Gynaecology, Polish Mother’s Memorial Health Institute, 93-338 Lodz, Poland; sylwestrzakoskarpatryk@gmail.com (O.S.); krekoram@poczta.onet.pl (M.K.); 3Department of Prenatal Cardiology, Polish Mother’s Memorial Hospital Research Institute, 93-338 Lodz, Poland; 4Department of Gynaecology and Obstetrics, Medical University of Lodz, 90-419 Lodz, Poland; 5Obstetrics Ward of the Polish Mother’s Memorial Health Institute, 93-338 Lodz, Poland; oszukowskip@gmail.com; 6Department of Obstetrics, Gynaecology and Gynaecology Oncology, Medical University of Lodz, 93-113 Lodz, Poland

**Keywords:** microbiota, dysbiosis, fetal heart defects, gut microbiome, congenital heart disease (CHD), metabolomics, prenatal health

## Abstract

Developments in medicine and biology in recent decades have led to a significant increase in our knowledge of the complex interactions between the microbiota and human health. In the context of perinatal medicine and neonatology, particular attention is being paid to the potential impact of the maternal microbiota on fetal development. Among the many aspects of this relationship, the question of the impact of dysbiosis on the development of fetal heart defects is an important one. In this article, we present an analysis of recent research and scientific evidence on the relationship between a pregnant woman’s microbiota and the development of fetal heart defects. We also discuss potential intervention strategies, including the role of probiotics and diet in optimising the maternal microbiota.

## 1. Introduction

Advances in medicine and biology in recent decades have greatly increased our knowledge of the complex interactions between microbiota and human health. In the context of perinatal medicine and neonatology, particular attention is being paid to the potential impact of the maternal microbiota on fetal development. Among the many aspects of this relationship, the study of the impact of dysbiosis on the development of fetal heart defects plays an important role.

Dysbiosis is an imbalance of the gut microflora. It is increasingly recognised as a risk factor for many chronic diseases. Understanding the mechanisms by which microbes living in the mother’s body can affect developing life is key to understanding and potentially minimising the risk of birth defects.

While interest in interactions between the microbiome and cardiovascular disease continues to grow, congenital heart disease (CHD) remains relatively understudied in this context. These patients are at increased risk of developing gut dysbiosis and intestinal disorders [1], highlighting the importance of the gut microbiome in patients’ clinical outcomes, including in the context of gut barrier function and inflammation, as demonstrated in previous studies reporting the association of a disturbed gut microbiome with adverse immunological outcomes in neonates with critical CHD [2]. In addition, cardiac surgery, especially those requiring extracorporeal circulation, can lead to a disturbed gut microbiome and gut barrier dysfunction, which, in turn, can induce systemic inflammatory responses after surgery [2].

The interplay between genetics and the prenatal environment, with a clear emphasis on the role of the mother’s gut microbiota, is crucial for fetal development and the risk of congenital heart defects. The neural crest cells that give rise to these processes not only shape neural structures but also contribute to the development of the immune system. This, in turn, has a direct impact on the formation of the thymus gland and the heart. Consequently, abnormal migration of these cells can lead to the development of CHD, which is particularly evident in conditions such as DiGeorge syndrome, where abnormal migration of neural crest cells is associated with defects in the heart and abnormalities in the development of the thymus and thus the immune system [3]. In this particular case, genetics are also not irrelevant. A study published in the Journal of the American College of Cardiology highlights the genetic links between heart and brain development, which is important for understanding the mechanisms leading to CHD. The genes responsible for the development of these organs are often shared, and their mutations can lead to complex developmental defects [4].

While the mother’s gut microbiota during pregnancy and its impact on the development of the fetal gut–brain axis have been widely discussed, its relationship to fetal heart development remains a relatively new topic. Recent studies have highlighted potential links between the microbiota and the fetal heart, showing that the maternal microbiota, while mainly focused on effects on the nervous system, also provides important clues about the cardiovascular system. Metabolites of the microbiota, such as short-chain fatty acids (SCFAs), play an important role in cell differentiation and the development of various systems in the fetal body, including the cardiovascular system. This suggests that the maternal microbiota may be a key factor shaping the cardiovascular health of the offspring [5].

In light of these findings, modulation of the maternal gut microbiota during pregnancy takes on new importance as a potential strategy for the prevention of CHD. Understanding the genetic links between heart and brain development and the mechanisms of microbiota–fetal communication opens new perspectives for preventive and therapeutic approaches. Such a multifaceted issue calls for increased research combining genetics, microbiology, embryology, and prenatal medicine.

In this article, we present an analysis of recent research and scientific evidence on the relationship between microbiota and gestational diabetes, cholestasis, and other health conditions that may affect fetal heart development. We also discuss potential intervention strategies, including the role of probiotics and diet in optimising the maternal microbiota (Figure 1).

## 2. Correlation Analysis and Research Perspectives

The microbiome is a complex system that interacts with every organ system in the human body. It contains approximately 100 trillion bacteria from more than 2000 species that coexist and are integral to homeostasis [6]. In a healthy body, Bacteroidetes and Firmicutes make up about 90% of the bacterial composition [7]. An imbalance of these bacteria, known as dysbiosis, leads to a decrease in healthy bacteria and an increase in pro-inflammatory bacteria.

Bacteroidetes and Firmicutes are key to maintaining gut health. One indicator of the health of the microbiome is the Firmicutes to Bacteroidetes (F/B) ratio. In infants, the F/B ratio is about 0.4, while in adults, it increases to about 10, indicating an increased proportion of Firmicutes in adulthood [8]. The higher the F/B ratio, the greater the correlation with obesity, coronary heart disease, stroke, heart failure, and autoimmune diseases [9,10].

The gut microbiome plays a key role in regulating the health and homeostasis of the body [11,12,13,14]. Dysbiosis, an imbalance of the gut microbiota, is a well-studied phenomenon in many disease states, particularly in the context of chronic antibiotic therapy [15,16,17] and in patient populations undergoing cardiac surgery [18,19,20], as well as in individuals with autoimmune diseases, inflammatory bowel disease, or obesity [9,14,15,21,22].

The above-mentioned conditions of an imbalanced microbiota lead to the development of epithelial barrier dysfunction (EBD) and, depending on the causative factor, this occurs through a mechanism of inadequate intestinal perfusion, hypoxia, impaired nutrition, or may be caused by reduced cardiac output due to heart failure. It is important to note that important contributors to EBD are intestinal infections, antibiotic overuse, physiological and psychological stress, and the effects of drugs such as non-steroidal anti-inflammatory drugs [23]. The effects of antibiotics on the gut microbiota can lead to a reduction in diversity, changes in microbial composition, and the emergence of antibiotic-resistant strains. This can have short-term consequences, such as antibiotic-associated diarrhoea, and long-term consequences, such as the development of obesity, allergies, or asthma. Newborns and young children are particularly vulnerable and may experience long-term changes in the gut microbiota after exposure to antibiotics. It can take up to 21 days to restore bacterial diversity after antibiotic treatment. Some individuals may experience a permanent decline in microbiome diversity [24].

Patients with congenital heart defects are at particular risk of developing dysbiosis and EBD due to a number of stressors, including abnormal intestinal blood flow, hypoxaemia, poor nutrition, and inferior cardiac performance, including valvular regurgitation and/or the presence of atrial or interventricular leaks [20,25]. In addition, in these patients, it is important to bear in mind chronic pharmacotherapy (including antibiotic or steroid therapy, especially in the perioperative period), which additionally overlaps with already existing EBD and promotes worsening changes in the gut microflora, which, in turn, may have a feedback and significant impact on the heart. Therefore, it seems important to continue research to better understand the complex interactions between dysbiosis and cardiac function, which may lead to the development of more effective therapeutic and preventive strategies for this particular patient population [25].

Decreased nitric oxide (NO) production in foetuses with heart defects often results in inadequate oxygenation of blood reaching the central nervous system (CNS). This decrease in NO can disrupt homeostasis and contribute to dysbiosis of the gut microbiota, exacerbating the vascular dysfunction that is central to the development of pulmonary hypertension (PHTN), which develops in the foetus secondary to congenital heart disease. Understanding the complex interaction between the gut microbiota, cardiac defects, and NO levels is essential for the development of more effective treatments and prevention [20].

Based on the available research, the relationship between diet and microbiota is significant. Dietary habits can influence gut microbiota composition, diversity, and inflammatory effects, especially high-protein and high-fibre diets, indicating a profound impact on gut health [26]. The gut microbiota is associated with inflammation and various body tissues. It varies from person to person and is influenced by diet and genetics, highlighting the need for further research to understand these interactions [27]. Furthermore, modulation of gut permeability by the gut microbiome has been linked to chronic inflammation and stress-related psychiatric disorders, highlighting the importance of microbiota interactions with barrier function and their impact on inflammation [28]. Understanding the impact of diet on the microbiome is key to developing strategies to maintain gut health and counter inflammation [29]. 

One very interesting and as yet incompletely understood corollary is the link between gestational cholestasis (ICP) and congenital heart defects. Gestational cholestasis is a condition affecting pregnant women, characterised by impaired bile flow and damage to liver cells, leading to elevated levels of bile acids in the blood. Studies suggest that gestational cholestasis may affect fetal heart function and contribute to the development of heart defects during fetal life [30].

Several mechanisms have been proposed that may explain how gestational cholestasis affects the fetal heart. One is that elevated levels of maternal bile acids, which cross the placenta, may directly affect the fetal myocardium, leading to cardiomyocyte dysfunction and affecting the cardiac stimulus-conduction system [31] by increasing NT-proBNP level, prolongation of PR interval length, and heart rate variability, resulting in the development of fetal arrhythmias and, in this mechanism, further adverse fetal myocardial remodelling and an increased risk of sudden fetal cardiac arrest [32]. 

It is worth noting that the exact mechanisms by which gestational cholestasis affects fetal heart development are not yet fully understood and require further research [31]. Nevertheless, monitoring and understanding fetal heart abnormalities in gestational cholestasis is crucial to provide appropriate care for both the mother and the unborn child [30]. Early detection, close monitoring, and timely intervention can help manage the potential risks associated with gestational cholestasis and improve outcomes for the foetus.

A study conducted at the University Children’s Hospital in Cairo found that 39.5% of infants with cholestasis had congenital heart defects (CHD), including leaky lesions, pulmonary valve stenosis, and complex defects [33]. Biliary atresia (BA), a common cause of cholestasis in infants, was associated with CHD in 35.9% of cases. In Alagille syndrome, the incidence of CHD was even higher at 93.7% [33]. Although cholestasis alone cannot directly cause congenital heart defects, cholestasis-related conditions such as Alagille syndrome can lead to both cholestasis and heart defects. It is important for clinicians to be aware of these potential links in order to provide comprehensive care for patients with cholestasis and congenital heart defects.

In the context of the above observations, our article aims to explore the potential mechanisms by which dysbiosis and changes in the pregnant woman’s microbiome may influence the development of fetal heart defects. By analysing the latest research and scientific evidence, we aim to identify key areas where therapeutic interventions, including optimisation of the maternal microbiota, may help reduce the risk of these serious developmental disorders.

## 3. Microbiome and Metabolomics in the Context of CHD and Beyond

Metabolomics provides valuable insight into congenital heart disease. It helps in diagnosing subtypes and predicting postoperative outcomes [34].

Metabolomics techniques, including nuclear magnetic resonance spectroscopy and mass spectrometry, enable detailed prenatal diagnosis of CHD and predictive analysis after heart surgery. Specific metabolomic fingerprints for CHD subtypes and markers of CHD-associated pulmonary arterial hypertension have been identified. Although promising, there is a need for more extensive research in this area. Metabolomic analysis of maternal amniotic fluid identified potential biomarkers for isolated CHD, revealing metabolites mainly involved in aldosterone synthesis, drug metabolism, and specific metabolic pathways.

Following previous discussions on the influence of maternal microflora on the development of fetal heart defects, it is worth highlighting new findings in the field of metabolomics. The research described in the article ‘Metabolomics: A New Tool in Our Understanding of Congenital Heart Disease’ [35] sheds light on the complex metabolic interactions that may be key to understanding this phenomenon.

The authors of this article highlight that CHD is associated with a variety of metabolic changes that are highly specific to certain types of defects [35]. For example, children with tetralogy of Fallot (TOF) showed significant changes in amino acid metabolism compared to children with other types of CHD. The present study primarily aimed to understand the role of the gut microbiota in CHD by comparing the gut microbiota of paediatric patients with TOF with that of healthy children. The study used stool samples and 16S rDNA gene sequencing to analyse differences in gut microbiota composition between the TOF group and the healthy children group. The analysis revealed differences in gut microbiota composition at the level of clusters and genera, identifying potential bacterial types as biomarkers to distinguish between healthy children and TOF patients. This study provides additional evidence of the potential impact of the maternal gut microbiota on CHD and highlights the need for further research into the role of the gut microbiome in CHD progression and outcomes. 

These findings are relevant when considering the impact of the maternal microbiota on fetal development. The gut microbiome is known to have a direct impact on host metabolism, including amino acid metabolism [36]. It is possible that dysbiosis, i.e., an imbalance in the intestinal microbiota, may lead to specific metabolic changes, which, in turn, influence the development of CHD in the foetus.

Further research is needed to understand exactly what mechanisms underlie these interactions. For example, is there a direct mechanism by which the maternal microbiota influences fetal metabolism, or is this influence indirect, related to the impact of the microbiota on maternal health and metabolic status?

The importance of metabolomics in understanding the impact of the mother’s microbiota on the development of fetal heart defects cannot be overstated. With further advances in technology and research, we will be able to better understand and manage these interactions, leading to better outcomes for both mother and baby.

The most common metabolic condition during pregnancy is diabetes mellitus (GDM). Studies show a significantly higher incidence of congenital heart defects in foetuses of women with diabetes compared to women without diabetes. Studies from France, Denmark, Saudi Arabia, and the United States present consistent results. Offspring of mothers with diabetes have an approximately four-fold increased risk of CHD compared to mothers without diabetes. Specific CHDs, such as atrioventricular septal defects, are more commonly associated with uncontrolled diabetes. Both type 1 and type 2 diabetes in mothers are associated with an increased risk of CHD in the offspring [37]. Changes in the composition of the gut microbiota, such as an increase in Firmicutes and a decrease in Bacteroidetes and Actinobacteria, have been observed in pregnancies with GDM [38], and the recent literature only highlights how much potential there is for microbiome-targeted therapies in gestational diabetes [39].

In another study titled ‘Association of maternal gut microbiota and plasma metabolism with congenital heart disease in offspring: a multi-omic analysis’, researchers carefully analysed the relationship between a mother’s gut microbiota and her child’s risk of congenital heart disease. This study provides valuable new information that may have implications for future preventive and therapeutic strategies [34].

In the study described here, several significant changes were observed in the gut microflora of mothers of infants with congenital heart defects compared to the control group. In particular, a significant decrease in microbial diversity was observed, which included lower levels of beneficial probiotics such as *Bifidobacterium* and *Lactobacillus*. These types are often associated with positive health outcomes and may contribute to intestinal barrier integrity, modulate immune function, and promote a healthy gut microbiota composition.

In particular, the genus *Bifidobacterium* has been found to have significantly reduced abundance in mothers of infants with CHD. The genus *Bifidobacterium* is usually considered beneficial and is often the target of probiotic treatment. *Streptococcus* was also found to be abundant and although some species in this genus may be pathogenic (such as *Streptococcus pneumoniae* or *Streptococcus pyogenes*), other species are commonly found in the healthy human microbiota or have probiotic properties (such as *Streptococcus thermophilus*). The context of the study did not indicate the pathogenicity of **Streptococcus** species with increased abundance, but given the association with CHD, it can be inferred that their prevalence does not indicate a positive impact on health in this particular scenario.

Regarding metabolites, the study performed untargeted metabolomics and identified significant differences in metabolic profiles between the two groups. Specific plasma metabolites that correlated with microbiological changes were involved in lipid metabolism pathways. For example, metabolites such as sphingomyelin (d18:1/18:0), acetyl carnitine and alpha-tocopherol (vitamin E) were associated with genera such as **Bifidobacterium** and **Streptococcus**, indicating that changes in the abundance of these gut microbes may relate to changes in maternal lipid metabolism, potentially affecting the development of CHD in the offspring [34].

The researchers hypothesised that the decrease in sphingomyelin in mothers of infants with CHD may be related to altered metabolic states of the mother, which may contribute to the risk of CHD development in the offspring. Sphingomyelin is known to play a role in signalling pathways that regulate insulin resistance, and disruptions in this pathway may affect embryonic cardiovascular development.

Based on the above results, it can be concluded that there are complex interactions between the maternal gut microbiota and plasma metabolites that may influence CHD risk in the offspring. The altered microbial populations and associated metabolites indicate a potential disruption of normal metabolic processes, possibly providing key information for future therapeutic interventions aimed at modulating the gut microbiota to prevent CHD.

Understanding which specific elements of the maternal microbiota and metabolome are associated with CHD may lead to new diagnostic and therapeutic strategies. It is possible that interventions to modify the maternal microbiota and metabolome, such as dietary changes or probiotic supplementation, may reduce the child’s risk of CHD.

These findings represent a step towards more personalised medicine, in which interventions are tailored to the individual characteristics of each pregnant woman’s microbiota and metabolome. Further research is needed to understand the mechanisms by which these changes affect fetal heart development and to develop effective intervention strategies. 

In the context of the above, it is worth noting the importance of the maternal microbiome, which is a key element in the transmission of microorganisms from mother to child. The importance of the microbiome in relation to health and disease is increasingly being recognised.

Another example is the knowledge that there is a type of resistant starch, i.e., a type of starch that is not digested in the upper gastrointestinal tract, meaning that it reaches the colon intact. There, it is fermented by the gut microbiota, leading to the production of short-chain fatty acids and other metabolites, which, in turn, provide health benefits [40]. 

Still, other studies have only confirmed that the mother’s microbiome can affect pregnancy and fetal development. For example, dysbiosis of the gut microbiome can lead to inflammation, which, in turn, can affect the duration of pregnancy and the birth weight of the baby [41]. Dysbiosis or imbalance in the gut microbiota can be caused by a number of factors, including diet, lifestyle, and stress. Inflammation in the mother’s body due to dysbiosis can lead to complications such as premature birth and low birth weight of the baby.

It has also been shown that specific bacterial strains in the maternal microbiome can influence the immune, metabolic, and neurological development of the child. For example, the presence of *Lactobacillus* and *Bifidobacterium* has been linked to beneficial effects on the child’s immune development, reducing the risk of allergies and asthma in later life [42].

In the context of these findings, understanding the mechanisms by which the maternal microbiome influences the health of the baby is crucial to developing interventions to optimise the maternal microbiome prenatally. This may include specific dietary advice, probiotic supplementation, or other strategies with which to promote healthy gut microflora.

Infections during pregnancy are often associated with a disruption of the mother’s microbiome. Changes in the mother’s microbiome, particularly in the gut, can affect her susceptibility to infection. For example, high levels of oestrogen and progesterone during pregnancy can increase a woman’s susceptibility to *Listeria monocytogenes* infection, which, in turn, can contribute to preterm birth or fetal death [43].

Changes in the gut microbiome are also observed physiologically during pregnancy. From the first to the third trimester of pregnancy, the relative abundance of proteobacteria increased from 0.73% to 3.2% in almost 70% of the pregnant women studied. At the same time, the abundance of actinomycetes increased from 5.1% to 9.3% in 57% of the women studied [44]. It is noteworthy that the species diversity of bacteria present in the pregnant woman’s body decreases in the third trimester compared to the first trimester.

*Bacteria* from the *Clostridiales* family were more numerous in the first trimester, while the number of bacteria from the *Enterobacteriaceae* and *Streptococcus* families increased in the third trimester [44].

Dysregulation of the maternal microbiome may be associated with several pathological conditions of pregnancy, including increased susceptibility to infection. Understanding how changes in the maternal microbiome may underlie pregnant women’s susceptibility to metabolic syndrome and infections may contribute to the development of preventive measures to improve pregnancy outcomes [45]. Understanding the dynamics of changes in the maternal microbiome during pregnancy and their impact on maternal and infant health is crucial.

For example, an increased abundance of Proteobacteria may be associated with inflammatory processes in the mother’s body. These processes may, in turn, affect the development of the foetus, increasing the risk of neurological, metabolic, and immunological disorders in the child’s later life. On the other hand, an increase in the abundance of *Actinobacteria* may be beneficial, as some of these bacteria produce antimicrobial and anti-inflammatory compounds.

In the context of infection, dysregulation of the maternal microbiome can increase susceptibility to infections such as *Listeria*, which are particularly dangerous for pregnant women and can lead to serious complications, including premature birth and fetal death [43].

In summary, changes in the maternal microbiome during pregnancy are complex and can have both positive and negative effects on maternal and child health. Understanding these changes and their consequences can lead to the development of interventions to optimise the maternal microbiome, which, in turn, can improve health outcomes for both mother and baby. However, further research is needed to fully understand these relationships and develop effective intervention strategies.

## 4. The Expanded Role of Probiotics in the Treatment of Heart Defects

Research into the use of probiotics in the treatment of heart defects is still at an early stage. However, there is evidence of a beneficial effect of *Bifidobacterium breve* on the gut microbiota of neonates with heart defects [46]. In the context of heart defects, mechanisms of action of probiotics may include modulation of the immune response, reduction of inflammatory processes, and protection against myocardial damage due to oxidative stress and inflammation [47].

Clinical and experimental studies have shown that probiotic supplementation can improve endothelial function, reduce oxidative stress, and modulate the immune response, which may have beneficial effects on cardiovascular health [48]. In the context of children with heart defects, further research is needed to understand how specific strains and combinations of probiotics can improve the health and quality of life of these patients.

## 5. The Role of Epigenetics

Another equally interesting area of research in the context of the influence of the maternal microbiota on the development of fetal heart defects is the role of epigenetics. Studies show that the maternal microbiome can influence the epigenetic profile of the foetus, which, in turn, can affect the heart and other aspects of prenatal development [23]. DNA methylation, gene regulation via non-coding RNA, and histone modifications are key epigenetic mechanisms influenced by maternal nutrition that are involved in cardiac development [20]. Nutritional interventions focusing on providing adequate micronutrients during the periconceptional period may reduce the incidence of CHD by affecting epigenetic modifications. Future research should focus on well-designed, controlled trials to better understand the correlation between maternal nutrition, epigenetic modifications, and CHD incidence [49].

## 6. Discussion

In the context of studying the influence of the maternal microbiota on the development of fetal heart defects, it is important to understand the role of probiotics and the gut microbiota in modulating immune responses and inflammatory processes. Studies have shown that probiotics, such as *Bifidobacterium* and *Lactobacillus*, can reduce the expression of pro-inflammatory genes and modify the gut microbiota [10,49,50].

Studies show that babies born with heart defects have an altered gut microflora, which may be associated with a risk of developing necrotising enterocolitis (NEC) [16]. NEC is a condition that mainly affects premature infants, but newborns with heart defects are also at risk. The use of probiotics, such as *Bifidobacterium longum*, has a beneficial effect on the gut microbiota of children with heart defects, suggesting a potential therapeutic pathway by which to improve their health [51]. 

Other reports suggest that gut dysbiosis, an imbalance in the gut microbiota, may influence the development and progression of cardiovascular disease. The gut microbiota and its metabolites, such as short-chain fatty acids, play a key role in modulating immune responses and inflammatory and metabolic processes that are closely linked to cardiovascular health [10].

SCFAs, such as butyric acid, propionic acid, and acetic acid, are produced in the gut as a result of fibre fermentation by intestinal bacteria. These fatty acids play a key role in modulating the immune response, meaning that they help regulate immune function. For example, they can reduce inflammation in the body, which is beneficial in inflammatory and autoimmune diseases.

Inflammatory and metabolic processes in the body are closely linked to cardiovascular health. For example, chronic inflammation can lead to the development of atherosclerosis and other cardiovascular diseases. SCFAs help regulate these processes, which can help reduce the risk of cardiovascular disease.

TMAO stands for trimethylamine N-oxide. It is a small molecule that occurs naturally in some foods, especially those rich in animal proteins. Initially, TMAO was thought to be harmless. However, recent studies suggest that elevated levels of TMAO in the blood can affect our health. High levels of TMAO are associated with an increased risk of certain diseases with a particular focus on cardiovascular disease including atherosclerosis and kidney disease [52]. TMAO is also produced by our body through the metabolism of dietary compounds such as choline, betaine, and carnitine. On the other hand, the recommended daily intake of choline for pregnant women is high at around 450–550 mg, which is crucial and positive for fetal brain development [53]. Despite potential concerns about TMAO production, choline remains an essential dietary component during pregnancy. To minimise the potential risk of negative effects of TMAO, a diet rich in probiotics is recommended, which can modulate the gut microbiota and potentially reduce TMAO production.

Probiotics, such as *Lactobacillus* and *Bifidobacterium*, have shown the ability to modulate the gut microbiota, which may contribute to reducing TMAO production [54]. These probiotic strains are known for their beneficial effects on gut health and their ability to promote a healthy gut microbiota profile. In addition, other probiotics, such as *Saccharomyces boulardii* and *Streptococcus thermophilus*, may also play a role in optimising the gut microbiota and reducing the risks associated with choline metabolism [55].

Understanding how the gut microbiota and its metabolites, such as SCFAs, affect cardiovascular health may lead to new treatment and prevention strategies in the context of children with heart defects. It is possible that manipulating the gut microbiota, for example, through the use of probiotics or prebiotics, could help reduce the risk or severity of heart defects and related complications [10].

In the context of heart defects, there is a need for a deeper understanding of how specific changes in the gut microbiota may influence the development and severity of these defects. It appears that interactions between the microbiota and the immune system may influence inflammatory processes and immune responses in cardiac tissue, which, in turn, may influence the development of cardiac defects [16] (Table 1).

With regard to the impact of the microbiota on the development of fetal heart disease, it is important to understand the role of choline, an essential nutrient for normal fetal brain development and function. Although choline is metabolised by the gut microbiota to trimethylamine (TMA), which is then converted in the liver to trimethylamine N-oxide (TMAO), a compound associated with an increased risk of cardiovascular disease, its role in neurodevelopment is essential [56].

In conclusion, while choline is a key nutrient in pregnancy and its intake is essential for fetal neurodevelopment, it is equally important to consider the potential impact of its metabolism on TMAO production. A probiotic-rich diet may be one way to mitigate this risk, ensuring the provision of essential nutrients for the health of both the mother and the developing foetus [57].

## 7. Conclusions

The development of research into the maternal microbiota and its impact on fetal health opens up new opportunities for medical and dietary interventions to improve maternal and child health outcomes. Understanding the mechanisms by which the maternal microbiota influences the development of fetal heart defects may lead to new diagnostic and therapeutic strategies. 

However, it is important that this approach is based on sound scientific evidence. Further research is needed to understand the complex interactions between the microbiota, genetics, diet, and other environmental factors and to assess the efficacy and safety of interventions that modulate the microbiota.

In an era of precision and personalised medicine, understanding individual differences in the microbiota and their impact on health can lead to personalised treatment and prevention strategies tailored to the unique needs of each mother and child.

## Figures and Tables

**Figure 1 biomedicines-12-00654-f001:**
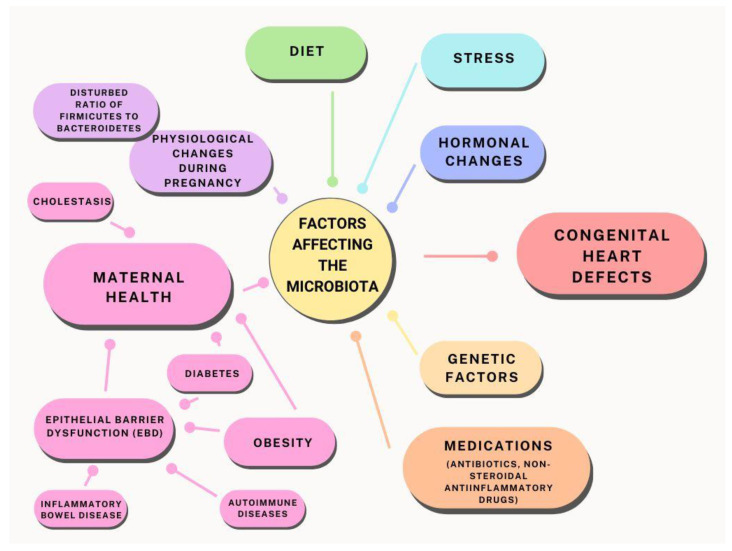
Relationship between maternal microbiota and factors influencing the potential development of heart defects.

**Table 1 biomedicines-12-00654-t001:** Influence of various factors on the development of congenital heart defects in the context of maternal microflora.

Factors Influencing the Development of Congenital Heart Defects (CHD)	Type of Impact	Description and Effects	References
Probiotics (e.g., Bifidobacterium, Lactobacillus)	Preventive	Improving gut health and microbiota balance can reduce CHD risk.	[46,50,51,54]
Dietary fibre and resistant starch	Preventive	Promotes healthy gut microflora, may reduce CHD risk.	[40,53]
Changes in amino acid metabolism	Variable	Some changes may be beneficial, others may contribute to the development of heart defects.	[34,35,36]
Changes in maternal metabolic profile	Variable	May affect development of CHD in the foetus; further studies needed.	[32,33,34]
Ratio of Firmicutes to Bacteroidetes	Indicator	Reflects health of microbiota; some profiles may increase CHD risk.	[8,10,38]
Physiological and psychological stress	Risks	This can lead to dysbiosis, potentially contributing to the development of CHD.	[41,44]
Antibiotic therapy	Risks	It can disrupt the gut microbiota, potentially contributing to the development of CHD.	[2,23]
Epithelial bowel barrier dysfunction (EBD)	Risks	This can lead to dysbiosis, potentially contributing to the development of CHD.	[23,25]
Autoimmune diseases and metabolic diseases (e.g., diabetes)	Variable	Impact depends on type of disease; may be associated with CHD risk.	[14,15]
Obesity and cardiovascular disease	Risks	Associated with an unfavourable microbiota profile, it can increase the risk of CHD.	[10,22]
Impaired nitric oxide (NO) production	Risks	It can contribute to dysbiosis and vascular dysfunction, influencing the development of CHD.	[20,41]

## Data Availability

All data refer to the relevant reference.

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
