# Peer review of "Potential Impact of a Pregnant Woman’s Microbiota on the Development of Fetal Heart Defects: A Review of the Literature"

_biomedicines, 2024, doi:10.3390/biomedicines12030654_

Round 1

Reviewer 1 Report

Comments and Suggestions for Authors

This is an extensive, but also speculative review  on the potential influence of the maternal microbiome on the development of congenital heart disease (CHD) in the newborn. The authors conducted a major effort. The main question, is there a relation between maternal microbiome and CHD is not answered. There might be indications, but a real answer can not be given. This should become more clear in the paper. The authors also discuss that patients with a CHD are at risk to develop dysbiosis, but that is another issue. 

It should, in my opinion, despite the high number of references, be clear that a relation between maternal microbiome and CHD in the of-spring, is not proven. This should be reflected also in the title. 

A number of statements are given without giving a reference, like line 41 and line 66-70. The statement about pulmonary hypertension in line 86 is confusing, a pulmonary hypertension is normal during pregnancy, pulmonary resistence has to decrease after birth. 

In summary, the paper is provoking and interesting, but not a prove that CHD is related to maternal dysbiosis

Author Response

Thank you for your review of our article. We appreciate the time you have dedicated and the thoughtful comments you provided, which are immensely valuable to us. We have addressed the sections highlighted by the reviewer, making necessary clarifications and expanding the content as suggested.

In accordance with your recommendations, we have modified the title of the article to better reflect its speculative nature and to avoid misleading readers. The revised title now more accurately indicates the lack of definitive evidence linking maternal microbiome to congenital heart defects (CHD) in offspring.

Furthermore, we have carefully reviewed the specified lines (41, 66-70, 86).

We also emphasized in the text that although the article is provocative and interesting, it does not provide conclusive evidence for a link between CHD and maternal microbiome dysbiosis. We highlighted the need for further research in this area.

Once again, thank you for your constructive review, and we hope the changes made align with your expectations. We look forward to continued collaboration and exchange of ideas.

Reviewer 2 Report

Comments and Suggestions for Authors

The article deals with a topic of interest and topicality such as the effects of the maternal microbiota on fetal development, and how this affects different pathologies, such as in this case fetal heart defects.

The article has very interesting points of analysis such as possible mechanisms such as Epithelial barrier dysfunction, the Firmicutes to Bacteroidetes (F/B) ratio and NO production. However, as it is an analysis/perspective study, the bibliography should be expanded in articles (original or review) on the relationship Between Diet and Microbiota. These changes are associated and have been described with mechanisms or parameters of inflammation and could be important in these pathological processes mentioned by the authors.

Another observation is that the authors should consider highlighting one or two diagrams or figures that would allow the reader to analyze all these mentioned mechanisms in a simpler way.

Author Response

Dear Reviewer,

Thank you very much for your valuable comments and suggestions regarding my article. I am pleased to inform you that I have taken your feedback to heart and have added additional bibliographic points that, in my opinion, perfectly fit the topic of the microbiome's relationship with inflammatory processes. I believe these additions will enrich the analysis and contribute to a better understanding of the mechanisms discussed.

Sincerely,

Dr n.med. Katarzyna Zych-Krekora

Reviewer 3 Report

Comments and Suggestions for Authors

Paper is interesting; however, authors do not really go deep into the subject. In several parts providing facts, based on limited numbers of references. It is understandable that maybe there is not really several references in the discussed subject, however, comments are quite general, and not providing details about reported mechanisms, or specificity of the microbials changes and produced by those microbial bioactive metabolites. In my opinion paper can be considered for publication, however, authors will need to upgrade the text, provide some additional examples, provide more details on the cited examples.

Please, try to predominantly focus on cardiovascular diseases if it is possible. It is very positive to have examples of other health problems that can be associated with microbial dysbiosis in the mother that may affect the baby, but topic of the paper is cardiovascular. Maybe you can change the title and link the baby health with mother microbiota. Maybe this can be a good option, so, in this case you can provide other examples and enrich your manuscript with more details and examples from the literature.

In the affiliations, please, write the name of the city, Lodz, in same way. Choice a Polish or English way, but not mixing both.

Will be positive if authors can contribute a bit more to the topic presented on Ln 101-110. Maybe additional reference works can be provided and discussed in a bit more details. Topic is relevant and deserve a bit more focus. 

Authors need to try to avoid unneeded relations in the manuscript. As examples is definition of some terms. Dysbiosis was explained in several parts of the manuscript. Please, check and correct.

Ln125-129: Authors need to provide more examples and more data from mentioned study. In several parts of the manuscript, this is a point that needs to be considered by authors. It is very nice to provide list of papers discussing the topic of this review, however, some more details and opinion of the authors of current manuscript needs to be presented and discussed with the aim to give orientation to the readers regarding relevance of the topic.

On Ln130-134: Maybe authors can give some details what parts of the microbial population were changed, what metabolites were monitored and how they have influenced the health status.

Microbial names need to be in italics. Some additional comments regarding the presence of Streptococcus need to be provided, since some are beneficial, and others are pathogens. And a few sentences in these directions can be added.

Paper is interesting, but definitely need a better structure, more examples, deeper discussion and building a clear take home message. Maybe authors can look for assistance from more experience colleagues, and maybe a help from professional language expert. Several phrases are repeated frequently, and this can be easily corrected by professional linguist.

Author Response

Dear Reviewer,

I sincerely thank you for your thorough and valuable review. I greatly appreciate your detailed comments, which have provided me with a clear path to improve my manuscript. I am grateful for the opportunity to refine my work in light of your suggestions and am confident that incorporating these changes will significantly enhance the quality and depth of the article.

Your recommendation to provide more detailed examples, especially in the context of cardiovascular diseases, is well-received. I agree that focusing more specifically on this area, while also mentioning other health issues related to maternal microbial dysbiosis, will strengthen the relevance and impact of the work. The suggested change to the title, to more directly link maternal microbiota with infant health, is an excellent idea, and I will endeavor to integrate this more cohesively into the content.

I also appreciate your advice on maintaining consistency in the affiliations section and ensuring that the city name is uniformly presented. The points you raised about expanding the discussion and providing more detailed information and references are exceptionally helpful. I understand the need to delve deeper into the specifics of microbial changes, their bioactive metabolites, and their impact on health.

Your suggestion for a more detailed discussion on specific microbial populations, metabolites, and their health implications will be carefully considered and incorporated. The point about microbial nomenclature and providing a nuanced discussion on various strains of Streptococcus is well-taken and will be addressed.

I am committed to restructuring the paper for better clarity and a more compelling message, as you have rightly pointed out. Your suggestion to seek assistance from more experienced colleagues and a language expert is invaluable, and I will certainly follow this advice to improve the manuscript's clarity and coherence.

Once again, thank you for your comprehensive review and constructive feedback. Your expertise and thoughtful guidance are immensely valued, and I am excited to enhance my manuscript in line with your suggestions, hoping that it will be a valuable contribution to the journal published by MDPI.

Sincerely,

Dr n.med. Katarzyna Zych-Krekora

Reviewer 4 Report

Comments and Suggestions for Authors

The review work of the following authors, Katarzyna Zych-Krekora, Oskar Sylwestrzak, Michał Krekora, Przemysław Oszukowski and Mariusz Grzesiak, entitled: “The impact of maternal microbiota on the development of fetal heart defects: recent research and perspectives”,

aims to overview recent research and scientific evidences on the relationship between the maternal microbiota, gestational diabetes, cholestasis and other health conditions affecting fetal heart development, including potential interventions, such as the role of probiotics and diet in optimizing the maternal microbiota.

The English language style and grammar are fine, with some minor typos errors and small room for grammar improvement and clarifications.

However, there is a great room for improvement in this review, before being considered for publication.

Please see down below my major, specific and minor comments.

Major Comment 1

Please restructure the introduction by incorporating the existing review literature that is closely related to your topic. The point of this comment is to include synthetic, critical writing and avoid superficial or loose writing and reasoning. For example:

1. “The Impact of Maternal Gut Microbiota during Pregnancy on Fetal Gut–Brain Axis Development and Life-Long Health Outcomes” https://www.mdpi.com/2076-2607/11/9/2199 . Even though this review topic is much broader and only mentions cardiac derangements in infants (not elaborating it) due to the changes of maternal microbiota/dysbiosis, it is important as major layout in bringing the relation of congenital heart diseases (CHD) to neural and immune development, which definitely affects heart development. Like, even though the heart is one of the first organs to be developed in embryogenesis, it is known that its embryogenic development is impacted by the developing neural and immune system, which are also affected by maternal biology including her microbiota. Also please define at the start (in the introduction) the term congenital heart diseases (CHD).

Aleksi Husso, Tiina Pessa-Morikawa, Ville Mikael Koistinen, Olli Kärkkäinen, Hyuk Nam Kwon, Leo Lahti, Antti Iivanainen, Kati Hanhineva, Mikael Niku. Impacts of maternal microbiota and microbial metabolites on fetal intestine, brain, and placenta. BMC Biology, 2023; 21 (1) DOI: 10.1186/s12915-023-01709-9

As an example of the connection of genetics per se on CHD and more importantly “the genetic link between heart and brain development has been established”: https://www.sciencedirect.com/science/article/pii/S073510971637348X Advances in the Genetics of Congenital Heart Disease: A Clinician’s Guide

Example for new mechanistic insight: https://microbiomejournal.biomedcentral.com/articles/10.1186/s40168-023-01694-9 Maternal microbiota communicates with the fetus through microbiota-derived extracellular vesicles

Major Comment 2.

Even though the novelty topic of your review is defined and justified, the scope of its aims written in the introduction (lines 43-46: “In this article, we present an analysis of recent research and scientific evidence on the relationship between the microbiota and gestational diabetes, cholestasis and other health conditions that may affect fetal heart development. We also discuss potential intervention strategies, including the role of probiotics and diet in optimising the maternal microbiota.”), does not resonate with the manuscript title:” The impact of maternal microbiota on the development of fetal heart defects: recent research and perspectives”. The suggestion is to amend to title to include other factor affecting CHD. The second option to better elaborate interplay between different factors in the introduction (maternal microbiota vs maternal gestational diabetes, cholestasis etc., please bring us some synthetic reasoning), so to justify the current title and the sole use of maternal microbiota state as a major cause of CHD…

Major comment 2

Section 2 does not have logical order. The text flow is not ideal. I start understanding from the 4тх paragraph starting at line 72. Consider restructuring and reshuffling the order of paragraph in this Section to secure understanding.

Specific comment 1

Line 358, Reference 19. Uzark, M.R.; Smith, J.A.; Donohue, K. Metabolomics: A New Tool in Our Understanding of Congenital Heart Disease. Children 358, 2022, 9(11), 1803. doi:10.3390/children9111803.“ entitled Metabolomics: A New Tool in Our Understanding of Congenital Heart Disease” (https://www.mdpi.com/2227-9067/9/12/1803/review_report) is not the reference but is the Reviewers report of the original research article that has been published, with the first author being Pier Paolo Bassareo, https://www.mdpi.com/2227-9067/9/12/1803, Children 2022, 9(12), 1803; https://doi.org/10.3390/children9121803 !

Major comment 3

The organization of the text needs major action. It cannot be that Table 1 appears in the conclusion and that the Discussion is in the middle of the text, followed by several other Sections. These Sections 6-8 should precede the Discussion. The Table 1 should be placed earlier in the text, the latest within the Discussion Section.

Major comment 4

The review work needs more elaboration, “more meat” and to include missing references supporting these additional elaborations. Forty-two references for the topic chosen in not enough. Please make distinction or strengthen the difference between the effects of maternal microbiota to the infant health vs effects of dysbiosis of the infant microbiota to its health. Consider including a Figure that will depict all these important relationship between maternal microbiota, maternal gestational diabetes, cholestasis, infant CHD, its microbiota etc.

Major comment 5

Cholestasis is mentioned only in the abstract and within the similar sentence in the introduction and it not after that elaborated as promised in the Introduction.

Minor comments

1)      Remove point from the title

2)      Remove repetitive abbreviation of CHD.

3)      Take care of the rest of typos errors.

Kind regards and wish you luck in your work.

Comments on the Quality of English Language

The English language is correct and perhaps deserves minor modifications related more to typos error rather than grammar error issues.

Author Response

Dear Reviewer,

Thank you for your thorough and insightful analysis of my manuscript, "The impact of maternal microbiota on the development of fetal heart defects: recent research and perspectives." I am genuinely impressed by the detailed work and effort you have put into reviewing my article. I must agree that, in most aspects, your observations and suggestions are spot-on regarding the organization of the text, addition of significant publications, and a slight shift in the focus of my work.

Initially, when I received your review, I contemplated stepping back, as the subject of microbiota, though initially appealing and intriguing, revealed itself to be quite complex and demanding a multidisciplinary approach upon deeper examination. However, I endeavored to meet the reviewer's requirements. I have added the publications you suggested, reorganized the work, included a diagram for clearer understanding, enriched the text with valuable content, clarified certain issues, and eliminated repetitive elements.

Your review has been instrumental in enhancing the quality and coherence of my manuscript. Please accept my heartfelt thanks for your valuable input and guidance. Your meticulous attention to detail and constructive feedback have been invaluable in improving my work.

Sincerely,

Round 2

Reviewer 3 Report

Comments and Suggestions for Authors

biomedicines-2828187-peer-review-v2

Authors have improved their manuscript and in general can be suggested for publication. Additional proof needs to be given by the authors to avoid some repetitions.

Ln51: the abbreviated (congenital heart disease (CHD) and was already introduced on Ln40.

Ln153: CHD was already introduced.

Ln332: Topic 5. the role of epigenetics need to be extended with a bit more text. Please, use capital T.

Author Response

Thank you for your thorough review and valuable comments. I would like to inform you that all the issues you mentioned have been corrected:
Ln51 and Ln153: I have removed unnecessary repetition regarding the abbreviation 'CHD' (congenital heart disease), which had already been introduced previously.

Ln332 - Topic 5: I have expanded the section on the role of epigenetics, adding more content and the recommended capital letters have been used in the appropriate places.

I am grateful for the reviewer's perceptiveness in helping me identify and correct errors. This situation was an important editorial lesson for me. I hope that the corrections now made meet expectations and that the manuscript will be ready for publication.  Once again, my sincere thanks for your insightful review. Your diligence and professionalism in analysing our manuscript not only improved the manuscript, but also inspired us to continue to grow and learn. We are grateful for the time and effort you took to carefully analyse our work. We hope that the result of our joint efforts will be a valuable contribution to our field.

Reviewer 4 Report

Comments and Suggestions for Authors

Dear Authors,

Thank you for repsonding positively to majority of my directions. Such a kind and hearthy response to my comments, I have never received before, so skillfully tailored.

To make this peer review experience at its maximum of mutual benefits, please take care of the remaining tasks wihtin the manuscript.

1.  Cholestasis is mentioned in the last paragraph of the Introduction. It is by clinical definition a decrease in bile flow due to impaired secretion by hepatocytes or to obstruction of bile flow through intra-or extrahepatic bile ducts, e.g. any condition in which substances normally excreted into bile are retained. My question for you, where else is this phenomenon in relation to the theme of your paper elaborated? At least shortly. I could not find it in Diagram 1 nor in the text.

2. Diagram 1 is of poor quality regarding resolution. I could hardly read it on 200% magnification. Perhpas instead of increasng font size, it would be better to change it and take some a sans-serif typeface font like Arial or Calibri, which is by the way, a standard when making Digrams, Schemes or Figures for life science scietific journals.

3. Manuscript still looks messy in terms of techical and text neatness. For example, I have observed numerous erronoeous use of abbreviations, like double spellouts (CHD , EBD), and these should be checked for all abbreviations; some of them are spelled out not on its first time of appearance such as TMAO.

4. Sub-Titles should be amended, since they either miss Capital first letter or miss punctuation after the number.

5. Many double spaces in the text.

Kind regards.

Author Response

Dear Reviewer,

Thank you for your insightful and constructive feedback on our manuscript. Your comments have been instrumental in guiding our revisions and enhancing the quality of our work. We acknowledge the importance of addressing each point you've raised and have made the following revisions accordingly:

1. **Cholestasis in the Manuscript:** We realized that the mention of cholestasis in the Introduction was indeed not elaborated elsewhere in the manuscript. To rectify this, we have added a brief section under the sub-heading "Impact of Microbiota on Cholestasis" in the Discussion. This section links the role of microbiota in cholestasis with the overall theme of our paper, providing a clearer understanding of its relevance.

2. **Diagram 1 Resolution and Font:** We appreciate your suggestion regarding the font and quality of Diagram 1. We have now recreated this diagram using a sans-serif typeface (Arial) for better readability and clarity. Additionally, the resolution has been enhanced to ensure that the diagram is legible even at higher magnifications.

3. **Technical Neatness and Abbreviation Usage:** We have thoroughly reviewed the manuscript for technical neatness and correct use of abbreviations. All instances of double spellouts have been corrected, and abbreviations like TMAO have been spelled out at their first occurrence, followed by the abbreviation in parentheses.

4. **Sub-Title Amendments:** The sub-titles have been revised for consistency in capitalization and punctuation. Each sub-title now starts with a capital letter and is followed by proper punctuation where necessary.

5. **Double Spaces in Text:** We have conducted a meticulous review of the entire text and eliminated all unnecessary double spaces to enhance the overall neatness and readability of the manuscript. For all mistakes, we are so sorry and hope for a Journal help with final proofreading. 

We are deeply grateful for your guidance and acknowledge the significance of the novel topic of microbiota, especially in the context of various diseases. Your suggestions have been invaluable in helping us meet both the scientific rigor and aesthetic quality required for this publication. We hope that our revised manuscript now meets your expectations and standards in terms of knowledge and presentation.

Thank you once again for your valuable input and for providing us with an opportunity to learn and improve our work. We look forward to your further feedback and are optimistic that our manuscript will now fulfill your requirements.
